# Using Optogenetics to Model Cellular Effects of Alzheimer’s Disease

**DOI:** 10.3390/ijms24054300

**Published:** 2023-02-21

**Authors:** Prabhat Tiwari, Nicholas S. Tolwinski

**Affiliations:** 1Department of Biochemistry and Molecular Biophysics, Kansas State University, Manhattan, KS 66506, USA; 2Division of Science, Yale-NUS College, Singapore 138527, Singapore; 3Program in Cancer and Stem Cell Biology, Duke-NUS Medical School, Singapore 169857, Singapore

**Keywords:** Alzheimer’s disease, amyloid, acetylcholinesterase, optogenetics

## Abstract

Across the world a dementia case is diagnosed every three seconds. Alzheimer’s disease (AD) causes 50–60% of these cases. The most prominent theory for AD correlates the deposition of amyloid beta (Aβ) with the onset of dementia. Whether Aβ is causative remains unclear due to findings such as the recently approved drug Aducanumab showing effective clearance of Aβ, but not improving cognition. New approaches for understanding Aβ function, are therefore necessary. Here we discuss the application of optogenetic techniques to gain insight into AD. Optogenetics, or genetically encoded, light-dependent on/off switches, provides precise spatiotemporal control to regulate cellular dynamics. This precise control over protein expression and oligomerization or aggregation could provide a better understanding of the etiology of AD.

## 1. Introduction

Although there have been several proposed causes of Alzheimer’s disease (AD), the amyloid hypothesis remains the most prominent [1]. Amyloid precursor protein (APP) is a type I transmembrane protein. APP is processed by sequential cleavage by either α-secretase and γ-secretase or β-secretase and γ-secretase. When cleaved by α and γ-secretase it yields APP intracellular domain (AICD) and P3 peptide. When cleaved by β and γ-secretase it yields AICD and amyloid beta (Aβ) peptide [2]. The Aβ peptide can exist as monomer, oligomer, and aggregate [3]. Aβ plaques are a hallmark of AD affected brains and can exist in intracellular or extracellular space [4,5,6,7]. Apart from Aβ accumulation, other key features of AD are tau aggregation, metabolic alteration, neuroinflammation, impaired neurotransmission, ER stress and signaling pathway alterations [8,9]. Tau aggregates or neurofibrillary tangles cause tauopathy in AD patients [10]. Aβ toxicity causes tau phosphorylation which destabilizes microtubules [11]. Phosphorylated tau aggregates and forms neurofibrillary tangles [12].

Aducanumab, an antibody-based immunotherapy for AD, has created controversy within the healthcare community. The drug’s ability to decrease Aβ concentration in the brain and curb neuronal toxicity has been shown in dose-dependent manner. Despite its promising results, the occurrence of Alzheimer’s-related imaging abnormality (ARIA) and hypersensitivity have raised concerns and questions about its safety [13]. Its effect on cognitive improvement has also been questioned [14]. There are however a series of new drugs in clinical trials offering hope for the future [15].

Several studies have shown metabolic alterations in AD prior to Aβ plaque formation [16,17]. Insulin and insulin like growth factor expression is altered in AD [18] and insulin resistance develops within the brain [19]. Additionally, there is a gene linked to both diabetes and AD, the insulin degrading enzyme [20]. With the commonality of symptoms between diabetes and AD, AD is also referred to as type 3 diabetes [19,20,21].

In addition to insulin sensitivity, symptoms of AD show changes in fatty acid and energy metabolism [16,22,23]. Aβ induces mitochondrial dysfunction affecting energy metabolism and generating oxidative stress contributing to neurodegeneration [24]. Both these effects further contribute to neuroinflammation [25,26]. But the cause-and-effect relationship between Aβ and mitochondria is not clear. Although Aβ and mitochondrial dysfunction correlate, a study in *C. elegans* shows that mitochondrial dysfunction precedes detectable Aβ accumulation [17].

There are many further aspects of AD such as alteration in Wnt, MAPK/ERK and insulin/Akt signaling [27,28,29]. Taken together, all the different aspects of AD are interconnected and complicate the biology of AD genesis and progression. Dissecting these aspects separately or in combination and studying their impact on tissues requires new models to develop a better understanding of AD. In this piece, we describe the application of published and new cell biological tools to study acute induction of the various proposed AD drivers. We discuss optogenetic Aβ, tau, and acetylcholinesterase (AChE) and the ways in which these tools could be applied to study AD. These optogenetics-based tools can provide a fundamental understanding of the biology of AD required for effective therapies.

## 2. Amyloid Cascade and AD

The amyloid cascade hypothesis states that Aβ monomers come together to form Aβ oligomers, which when secreted, form plaques (Figure 1). AD patient brains show aggregation of Aβ into plaques but this finding is mainly a correlation and whether these plaques are cause or consequence is not trivial to establish [5]. Plaques are the most obvious form of Aβ observed, but recent studies suggest that Aβ oligomers are the key effectors in AD progression [3,6,7].

Neurons in the brains of patients and in animal models show intracellular accumulation of Aβ [30]. Intracellular accumulation of Aβ can occur either by intracellular formation or by cellular uptake of extracellular Aβ. Amyloid deposition does not always correlate with cognitive decline as one study found similar levels of amyloid in symptomatic and asymptomatic individuals, but in a small sample size [31].

These finding suggest that the correlation between plaque reduction and its effect on cognitive decline will require more studies, and that there are still some challenges to be resolved for this hypothesis [32,33]. For example, what is the effect of acute Aβ aggregation or oligomer formation. Optogenetic based spatiotemporal control of Aβ aggregation has provided a tool to understand the effects of acute aggregation [34,35].

## 3. Tauopathy in AD

Aβ toxicity has also been linked to tau aggregation leading to tauopathy. Tau is a microtubule binding protein abundant in neuronal cells important for the stabilization of microtubules. Tau proteins tend to aggregate in response to mutations and posttranslational modifications (Figure 1). Tau hyperphosphorylation is a posttranslational modification causing tau aggregation. Hyperphosphorylated tau destabilizes microtubules affecting axonal transport leading to neurodegeneration [36].

Alternative splicing of tau results in multiple isoforms that contain either 3 or 4 microtubule binding repeats. Normally isoforms with either 3 or 4 repeats are present in 1:1 ratio but in AD this ratio is altered to 1:2 [10]. Aβ toxicity has been shown to cause hyperphosphorylation of tau linking tau to Aβ [11].

Studying forms of tauopathy that are independent of Aβ toxicity can shed light on AD progression and will help in dissecting the Aβ cascade into discrete units. For example, cases where amyloid plaque clearance does not improve cognition in AD might be due to the downstream effects of Aβ cascade. Understanding the acute Aβ aggregation effects in terms of the immediate and late pathophysiology are a key to understanding the Aβ cascade.

## 4. Metabolic Alteration and AD

Metabolic changes in AD patients have been reported [23] and show correlation with increased Aβ accumulation [37]. AD shows signatures of metabolic disorders and thus can be considered as a metabolic disease [16]. AD shares molecular and biochemical similarity to the type 1 and type 2 diabetes mellitus. There is impaired glucose utilization, insulin response, energy metabolism, and insulin resistance in AD patients [19,20,21,38].

AD patients show altered energetics in neurons and glial cells [39]. Reactive oxygen species (ROS) levels increase caused by high oxidative phosphorylation in mitochondria linked to increased Aβ toxicity increasing oxidative stress [16,24,40]. Further reported effects include alteration in glucose metabolism, glycolysis, tricarboxylic acid (TCA) cycle, oxidative phosphorylation, and the pentose phosphate pathway [41].

Positron emission tomography imaging shows correlation with glucose metabolic dysfunction and AD [42]. Insulin treatment has been shown to improve memory in AD patients [43]. Metabolic effects have been observed in an optogenetic model of Aβ aggregation in *Drosophila melanogaster* and *C. elegans* [34]. Metabolic effects can precede detectable Aβ aggregates suggesting it as a causative factor in Aβ pathology and can be ameliorated with the anti-diabetic drug metformin [17]. In model organisms, measuring metabolism in response to acute aggregate formation will lead to a better understanding of Aβ’s metabolic effects.

## 5. Cholinergic Hypothesis

The cholinergic hypothesis of AD states that the degeneration of cholinergic neurons and the reduction in acetylcholine (ACh) levels in the brain are related to the development of AD. ACh is synthesized in the cholinergic neurons by the enzyme choline acetyltransferase and transported to the synaptic vesicles by the vesicular acetylcholine transporter. ACh plays a crucial role in cognitive function, memory, attention, learning, and other physiological processes. The degeneration of cholinergic neurons, which is found in AD, causes alterations in cognitive function and memory loss. Other factors, such as the reduction in nicotinic and muscarinic ACh receptors, a deficit in excitatory amino acid neurotransmission, and the use of cholinergic receptor antagonists, also contribute to the progression of AD. The cholinergic hypothesis is supported by evidence of reduced presynaptic cholinergic markers in the cerebral cortex, severe neurodegeneration of the nucleus basalis of Meynert in the basal forebrain, and the ability of cholinergic agonists to reverse the effects of cholinergic antagonists on memory decline [8,44,45,46,47,48].

AChE is an enzyme that hydrolyses the ACh to acetic acid and choline to check its further dispersal [49]. It is associated with Aβ plaques [50]. AChE is present in monomeric, dimeric, and tetrameric forms with an increase of monomeric forms observed in AD [51]. AChE inhibitors are used to treat neuropsychiatric symptoms of AD but they tend to lose effectiveness over time and have side effects [52]. To understand the effect of AChE inhibition in AD and its effect on Aβ pathophysiology, an optogenetics approach could be used. Several photoswitchable kinase models have been made by using optogenetic tools [53,54,55]. Here we propose an optogenetic model for AChE inhibition (Figure 2). This optogenetic model is dependent on photodimerizable-DRONPA (pdDronpa), a photosensitive protein that dimerizes upon exposure to 400 nm light and thus cages the active site of the enzyme. Dimerization is reversed upon 500 nm light exposure making the enzyme functional. pdDronpa has been previously used in generating photoswitchable MAPK/ERK Kinase (MEK) [53].

## 6. Endoplasmic Reticulum Stress

Another reported driver of AD is endoplasmic reticulum (ER) stress, which refers to the accumulation of unfolded or misfolded proteins within the ER, leading to disruptions in cellular homeostasis. There is increasing evidence that ER stress contributes to the development and progression of AD [9].

The ER stress and errors in protein folding are associated with aging. It is also associated with the neuroinflammation [56] and immunosuppression [57]. Understanding the role of ER stress in AD pathogenesis may provide new opportunities for treatment helping to relieve cognitive dementia symptoms.

## 7. Inflammation and AD

Alterations in pathways related to oxidative stress are associated with inflammation [23,26]. Normal inflammation functions as a defense mechanism against toxins and pathogenic infections. But sustained inflammation, a major factor in AD, is damaging to the survival of the cells and tissues. It has been shown to exacerbate AD pathophysiology. Activated microglia accumulate around the Aβ plaque in patients [58], and are associated with both Aβ and tau [59]. Other glia such as astrocytes and oligodendrocytes also contribute to AD pathophysiology [26,60].

Glutathione a major antioxidant with anti-inflammatory effects shows positive correlation of its pathway metabolites with AD [23]. Certain polyunsaturated fatty acids are known to have anti-inflammatory effects, for example the epoxy fatty acid has been shown to have anti-inflammatory activity in the mouse model of AD. Inhibiting the hydrolysis of epoxy fatty acid delays AD progression [61,62].

In an optogenetic *Drosophila* model of AD acute Aβ aggregation shows inflammation [63]. Anti-inflammatory compounds have been considered as potential therapeutic agents in AD [64]. The optogenetic model of acute Aβ aggregation can be used to study the effect of anti-inflammatory factors on AD pathophysiology.

## 8. Amyloid-β Interaction with Signaling Pathways

One, perhaps understudied aspect of AD is the impact that it has on signaling pathways. Many developmental signaling pathways change roles during adulthood to regulate aspects of homeostasis. Apart from mutations that lead to cancer, less drastic changes can affect a variety of metabolic and other processes in the adult. In AD, MEK/ERK and mTOR pathway activation are detrimental while Wnt pathway activation is beneficial [65,66,67,68].

One factor that suggests hope for future AD treatments is that activating Wnt and inhibiting both ERK and mTOR show amelioration of AD symptoms [69]. The combination treatment shows lifespan extension in model organisms and could be applied as a combination treatment for AD [70]. Activation of Wnt and inhibition of MEK with a two-drug combination is known as a pro-stem cell treatment suggesting that this may function as a cellular rejuvenation process similar to cell rejuvenation treatments using Yamanaka factors [71,72]. These treatments ameliorate various hallmarks of aging and extend lifespan suggesting that cells can be reprogrammed to a younger state perhaps leading to better misfolded protein response, amelioration of AD symptoms, and perhaps reversing cognitive decline.

## 9. Optogenetics as a Method to Study the Effect of Aβ Aggregation

Updating current models for understanding of AD biology should be a priority as new advances in cell biological methods have accelerated in recent years. Genome editing has become routine, live imaging with a rainbow of fluorescent proteins is accessible, and optogenetics allows protein manipulation in a spatial and temporal manner [73,74]. For example, to study the initial steps of Aβ oligomerization optogenetic Aβ could be used. Optogenetics can be used to regulate expression and oligomerization of proteins. This tool has been applied to various model organisms with many tools developed. One popular tool uses cryptochrome2 (CRY2) protein based optogenetic constructs. CRY2 is a phytochrome from *Arabidopsis thaliana*. CRY2 oligomerizes quickly in presence of blue light (488 nm) and oligomerization is reversed when blue light is turned off. This can be used to investigate the protein aggregation-based diseases. The photolyase homology region (PHR) of CRY2 is responsible for the oligomerization. The PHR region can be fused to the protein of interest under investigation (targeted for oligomerization or aggregation) and a fluorescent protein to follow the expression, localization, and aggregation of the protein of interest through live imaging.

The optogenetics based AD models in *Drosophila,* zebrafish and *C. elegans* use acute Aβ aggregation [34,63]. They utilize a CRY2-Aβ-mCherry fusion construct which shows aggregation when exposed to the blue light. The models show normal development when reared in the dark but display neurodegenerative defects when exposed to blue light. This model when applied to other, non-aggregating proteins, shows reversal of aggregates (Epidermal Growth Factor Receptor, β-Catenin in both embryonic and adult models) when kept in dark after blue light exposure [75,76,77]. This reversal is not observed with the CRY2-Aβ-mCherry due to the aggregating nature of the Aβ peptide, but it provides a model to study effects of acute Aβ aggregation. *Drosophila* embryos expressing CRY2-Aβ-mCherry exposed to specific amounts of blue light show significant neuronal defects. The sensitivity to light levels provides the flexibility to modulate the severity aggregation and therefore severity of phenotype by modulating light induction intensity [76].

The model can be further extended to test one still unresolved issue concerning the effect of intracellular vs. extracellular Aβ aggregates especially with recent evidence pointing to the significance of intracellular Aβ aggregates [6,30]. By adding a secretion signal sequence to the CRY2-Aβ-mCherry fusion construct, distinct effects of intracellular and extracellular Aβ aggregates can be studied (Figure 3). This optogenetic system has already been applied to test the effects of signaling pathway activation and drug treatment on Aβ aggregation [34,63].

## 10. Opto-Tau

Tau hyperphosphorylation and formation of neurofibrillary tangles are further features of AD progression that can be modelled optogenetically. Aβ toxicity promotes tau NFT formation, which resemble aggregates. Applying the same basic CRY2-mCherry technique but fused to tau should lead to NFT formation in a light inducible manner. CRY2-Tau-mCherry can be induced to aggregate by exposure to blue light (Figure 4A). A second approach would be to optogenetically activate kinases that cause tau-hyperphosphorylation like Glycogen Synthase Kinase 3 (GSK-3β), Extracellular signal Regulated Kinase (ERK), Cyclin Dependent Kinase (CDK5) [78]. Another light sensitive protein domain, Light Oxygen Voltage Sensing domain (LOV), has been used in various optogenetic strategies. Vivid (VVD), a LOV domain protein from *Neurospora crassa*, homodimerizes in response to blue light, has been used to generate light regulated kinase switches [55]. Designing the tau-hyperphosphorylating kinases with the VVD will provide optogenetic control for tau-hyperphosphorylation and provide an alternative way of generating tau-NFT (Figure 4B). The differences in the metabolic profile, neuroinflammation and neurodegeneration between CRY2-Aβ-mCherry and CRY2-Tau-mCherry will shed light on the distinct pathophysiology of Aβ and tau aggregates in AD. Moreover, these will provide a model to test tau specific interventions.

## 11. Conclusions

The global healthcare system is facing a major challenge with the rapid increase in the number of Alzheimer’s disease cases. The current therapeutic approaches have failed to provide an effective cure, making it imperative to find new and innovative methods for the prevention, treatment, and diagnosis of the disease. In this piece we focused on a variety of cell biological tools for basic research into the molecular effects of the disease, but many other possible approaches will be necessary. For example, precision medicine offers a personalized approach that considers the individual’s genetic, environmental and lifestyle factors, leading to the development of tailored therapies. Precision medicine initiatives are being launched globally to integrate personalized models with clinical medicine and will prove crucial for AD treatment [79].

Aβ oligomers and plaques correlate with cellular symptoms like metabolic alteration, neuroinflammation and signaling pathway modulations. As these effects appear to be interrelated, dissecting the mechanisms of Aβ based pathophysiology is complicated. Modern tools of cell biology like optogenetics offer precise control over aggregation, offer the ability to vary the extent of aggregation and may provide a handle to dissect Aβ and tau biology in AD. These tools may provide the means to address questions such as where Aβ functions intracellular vs extracellular, is the Aβ oligomer or aggregate responsible, and compare the effect of tau-NFT to Aβ.

The combination of new approaches to basic AD research, the testing of new therapies, personalized medicine, cellular rejuvenation, and many other avenues yet to be explored will yield effective therapeutic regimens for this complex and devastating condition.

## Figures and Tables

**Figure 1 ijms-24-04300-f001:**
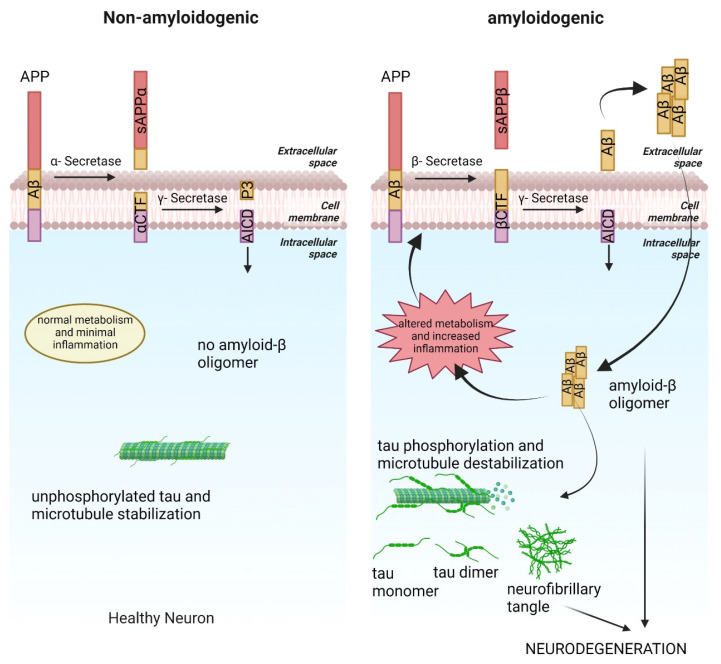
Differences between non-amyloidogenic and amyloidogenic pathways in Alzheimer’s disease (AD). Non-amyloidogenic cells have normal cellular metabolism, a low level of inflammation, and stable microtubules. Amyloidogenic cells show metabolic alteration, increased inflammation, amyloid aggregates, destabilized microtubules, and tau-aggregates.

**Figure 2 ijms-24-04300-f002:**
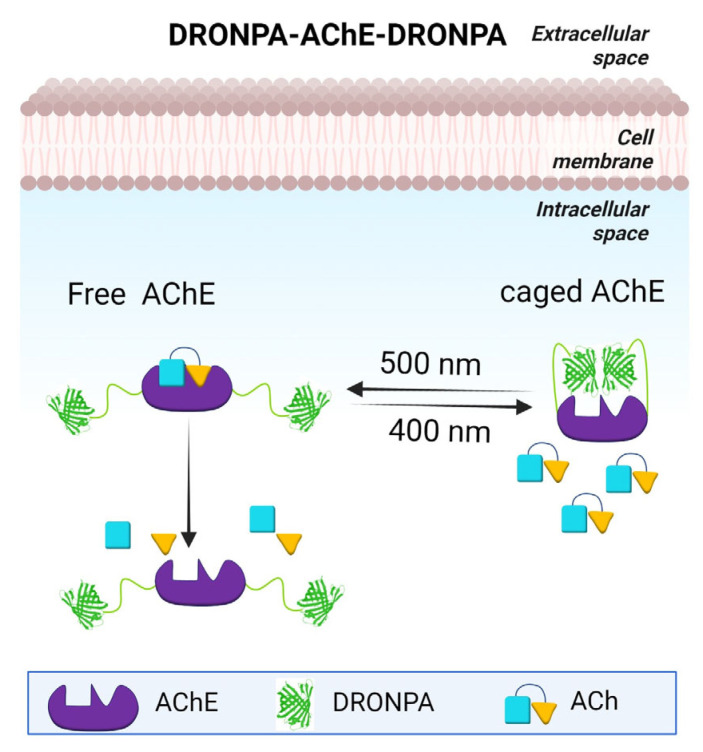
Optogentic control of acetylcholinesterase (AChE). Activity of AChE could be regulated by light with 400 nm light activating DRONPA dimerization blocking the enzyme’s active site. 500 nm light reverses the dimerization allowing AChE to hydrolyse Acetylcholine into acetic acid and choline.

**Figure 3 ijms-24-04300-f003:**
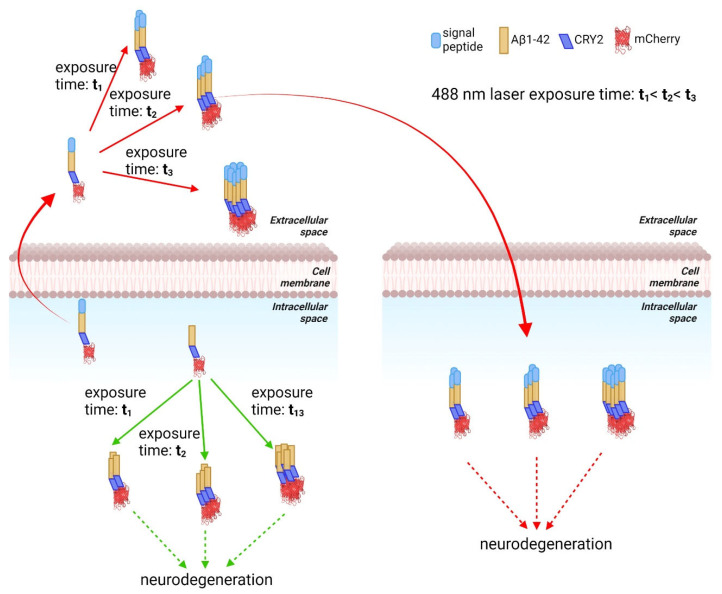
An optogenetic strategy to vary levels of amyloid in intracellular and extracellular space. Cryptochrome 2 (Cry2)-mCherry Aβ aggregates upon exposure to 488 nm laser light. By controlling the exposure time, the level of aggregation can be varied. A signal peptide for secretion to extracellular space can be added to CRY2-mCherry Aβ. Neighboring cells can import the then secreted Aβ.

**Figure 4 ijms-24-04300-f004:**
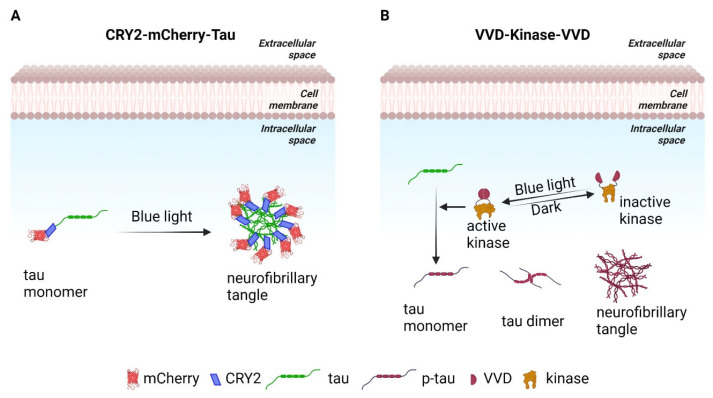
Strategies for making tau-aggregates. (**A**) A construct of CRY2-mCherry-tau that will respond to blue light to form tau aggregates. (**B**) A light activated kinase to phosphorylate tau forming aggregates.

## Data Availability

Not applicable.

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
