# Peer review of "Using Optogenetics to Model Cellular Effects of Alzheimer’s Disease"

_ijms, 2023, doi:10.3390/ijms24054300_

Round 1

Reviewer 1 Report

The manuscript entitled “Modeling downstream cellular effects of acute amyloid-beta aggregation” represents an opinion on new approaches for understanding Aβ function. The authors discuss the application of optogenetic techniques to gain insight into AD and provide a better understanding of its etiology. The manuscript is interesting and well-prepared. It fits well with the scope of the Journal. This topic is highly important. Still, I have some major concerns.

The authors should include the discussion of AChE involvement in AD since it is inevitable (please, see https://doi.org/10.2174/0929867324666170705123509, 0.2174/0929867330666230112160522). Besides, most drugs for AD treatment are based on AChE inhibition. It has to be thoroughly discussed.

Also, the evidence suggests that AD is associated with the induction of endoplasmic reticulum stress and maladaptive unfolded protein response. It should be mentioned and discussed. Please, see http://dx.doi.org/10.1038/s41419-022-05153-5.

Author Response

The manuscript entitled “Modeling downstream cellular effects of acute amyloid-beta aggregation” represents an opinion on new approaches for understanding Aβ function. The authors discuss the application of optogenetic techniques to gain insight into AD and provide a better understanding of its etiology. The manuscript is interesting and well-prepared. It fits well with the scope of the Journal. This topic is highly important. Still, I have some major concerns.

The authors should include the discussion of AChE involvement in AD since it is inevitable (please, see https://doi.org/10.2174/0929867324666170705123509, 0.2174/0929867330666230112160522). Besides, most drugs for AD treatment are based on AChE inhibition. It has to be thoroughly discussed.

We have added a paragraph on the Cholinergic Hypothesis.  We have also suggested an optogenetic approach. 

Also, the evidence suggests that AD is associated with the induction of endoplasmic reticulum stress and maladaptive unfolded protein response. It should be mentioned and discussed. Please, see http://dx.doi.org/10.1038/s41419-022-05153-5.

We have added a paragraph about ER stress. 

Reviewer 2 Report

Review article titled "Modeling downstream cellular effects of acute amyloid-beta aggregation" by Tiwari AND Tolwinski  DESCRIBES THE CELLULAR EFFECTS of aggregation of Ab and modeling for its downstream effects. They summarized the topic in nice and easy diagrams. I appreciate this humble and simple trial to discuss this important point.

Some parts are overstated or exaggerated such as abstract "Across the world a dementia case is diagnosed every three seconds" what was the source of this serious statement!???

Also " Alzheimer’s disease 8 (AD) causes 50-60% of these cases with current estimates standing at fifty-five million cases world- 9 wide, with an economic impact estimated at $1.3 trillion."

Title: need to be amended to express the content : what are the downstreams? ". Optogenetics, or genetically encoded, light-dependent on/off switches, provides precise spatiotemporal control to regulate cellular dynamics" can this part included in title by some modifications.

I think section "4. Metabolic alteration and AD" can come before section 2.

Conclusion is not useful & did not summarize the important points or mention the future directions & need to be rewritten.

Author Response

Review article titled "Modeling downstream cellular effects of acute amyloid-beta aggregation" by Tiwari AND Tolwinski  DESCRIBES THE CELLULAR EFFECTS of aggregation of Ab and modeling for its downstream effects. They summarized the topic in nice and easy diagrams. I appreciate this humble and simple trial to discuss this important point.

Some parts are overstated or exaggerated such as abstract "Across the world a dementia case is diagnosed every three seconds" what was the source of this serious statement!???

This comes from the Alzheimer’s Association and their presentation on the economic impact of AD. As we don’t discuss this further it has been removed.

Also " Alzheimer’s disease 8 (AD) causes 50-60% of these cases with current estimates standing at fifty-five million cases world- 9 wide, with an economic impact estimated at $1.3 trillion."

As above.

Title: need to be amended to express the content : what are the downstreams? ". Optogenetics, or genetically encoded, light-dependent on/off switches, provides precise spatiotemporal control to regulate cellular dynamics" can this part included in title by some modifications.

We have changed the title to reflect the reviewer's comments.

I think section "4. Metabolic alteration and AD" can come before section 2.

There has been substantial reorganization due to the referees’ comments.

Conclusion is not useful & did not summarize the important points or mention the future directions & need to be rewritten.

We have rewritten this.

Reviewer 3 Report

Comments:

In the present opinion article, the authors assessed the implications of acute amyloid beta deposition in the onset of neurological disorders. The topic is relevant and interesting, but the paper design, the data provided, and the organisation of information still need changes before they become adequate. Significant requirements are listed below in order to improve the initial form:

1.     The manuscript needs to be revised in terms of the use of abbreviations. There are abbreviations not explained in the text (e.g., L12-FDA, L96-ROS, L99-TCA etc.). Abbreviations should be defined the first time they appear (added in brackets after the written-out form, according to the instructions for authors), separately in each of the three sections: the abstract; the main text; and under the first figure or table. When defined for the first time, the abbreviation should be added in parentheses after the written-out form Please revise and check all the abbreviations to be explained at their first mention in the text.

2.     Please revise the manuscript in terms of editing, informal language and written form (L 40 insulin should be written in small letters etc.).

3.     Since aducanumab has been referred to only in the abstract with no other references in the main text, it is advisable to briefly present in the introduction some data about the safety and efficacy profile of aducanumab in relation to AD. I suggest checking and referring to: PMID: 35231697.

4.     Figure 1 is not announced in the main text. Please revise this aspect.

5.     Aim of the study is poorly described.  As the topic is not a new one, in the last paragraph of Introduction, please highlight/detail the special aspects/novelty that your work brings to the field, and what differentiate this paper from other in the same topic.

6.     Chapters 2,3,4,5,8 are organised as very long paragraphs that are hard to read/understand. Please revise and restructure them into smaller, easy to read paragraphs.

7.     Beyond the conventional form of the disease, tau diseases have a wide range of manifestations. It is advisable in Chapter 3 to briefly present these manifestations according to the pathology.

8.     In L103 it is not clear whether the authors refer to the species Drosophila melanogaster or to the genus Drosophila, then they must refer to Drosophila spp. Please revise this issue.

9.     It is especially important in the context of AD being an incurable disease to briefly address in the manuscript the concept of precision medicine in the management of AD. I suggest checking and referring to: PMID: 35780617.

10.  The current shortcomings and how they could be addressed in future research directions should be better highlighted in the conclusions section.

Author Response

In the present opinion article, the authors assessed the implications of acute amyloid beta deposition in the onset of neurological disorders. The topic is relevant and interesting, but the paper design, the data provided, and the organisation of information still need changes before they become adequate. Significant requirements are listed below in order to improve the initial form:

  1. The manuscript needs to be revised in terms of the use of abbreviations. There are abbreviations not explained in the text (e.g., L12-FDA, L96-ROS, L99-TCA etc.). Abbreviations should be defined the first time they appear (added in brackets after the written-out form, according to the instructions for authors), separately in each of the three sections: the abstract; the main text; and under the first figure or table. When defined for the first time, the abbreviation should be added in parentheses after the written-out form Please revise and check all the abbreviations to be explained at their first mention in the text.

We have fixed these errors. 

  1. Please revise the manuscript in terms of editing, informal language and written form (L 40 insulin should be written in small letters etc.).

These have been changed.

  1. Since aducanumab has been referred to only in the abstract with no other references in the main text, it is advisable to briefly present in the introduction some data about the safety and efficacy profile of aducanumab in relation to AD. I suggest checking and referring to: PMID: 35231697.

We have added a paragraph on this.

  1. Figure 1 is not announced in the main text. Please revise this aspect.

Apologies for this oversight.

  1. Aim of the study is poorly described.  As the topic is not a new one, in the last paragraph of Introduction, please highlight/detail the special aspects/novelty that your work brings to the field, and what differentiate this paper from other in the same topic.

We have added this to the introduction.

  1. Chapters 2,3,4,5,8 are organised as very long paragraphs that are hard to read/understand. Please revise and restructure them into smaller, easy to read paragraphs.

We have revised this.

  1. Beyond the conventional form of the disease, tau diseases have a wide range of manifestations. It is advisable in Chapter 3 to briefly present these manifestations according to the pathology.

This has been added.

  1. In L103 it is not clear whether the authors refer to the species Drosophila melanogaster or to the genus Drosophila, then they must refer to Drosophila spp. Please revise this issue.

This has been fixed.

  1. It is especially important in the context of AD being an incurable disease to briefly address in the manuscript the concept of precision medicine in the management of AD. I suggest checking and referring to: PMID: 35780617.

This has been added.

  1. The current shortcomings and how they could be addressed in future research directions should be better highlighted in the conclusions section.

The conclusion has been rewritten to reflect this.

Round 2

Reviewer 1 Report

The authors addressed all my comments. I recommend this manuscript for publication in its present form. 

Author Response

We thank the reviewer for the kind comments.  

Reviewer 2 Report

thx

Author Response

Thank you!

Reviewer 3 Report

The authors changed the type of the manuscript in Review. Along with this, the requirements also change. Few of my major concerns:

L73. You cannot propose, as the manuscripts you are referring to in a Review are already published.

L74. "We discuss.." appear twice.

No novelty of the paper is revealed in the aim of the study.

Please check the Instructions for authors regarding Abbreviations and apply those requests to your figures as well.

All the sections must be numbered in all types of manuscript, including for Review.

A 2nd section Methodology for literature searching must be developed.

Tabulated part is missing.

Information is much too poor for a Review manuscript.

Too many empty lines. What for?

Less than 9 pages of real manuscript, in MDPI format which is 2/3 pages occupied with text cannot be considered a Review.

Author Response

The authors changed the type of the manuscript in Review. Along with this, the requirements also change. Few of my major concerns:

We apologize for the error; it was accidental during the online submission process.  This was always meant as an Opinion piece about Optogenetics and not a comprehensive review of AD. This has been corrected by the editor in the current version.

L73. You cannot propose, as the manuscripts you are referring to in a Review are already published.

Corrected.

L74. "We discuss.." appear twice.

Corrected.

No novelty of the paper is revealed in the aim of the study.

The study discusses possible optogenetic tools to be made. Not sure what the reviewer is requesting here. 

Please check the Instructions for authors regarding Abbreviations and apply those requests to your figures as well.

All of the abbreviations used are explained in the text now.

All the sections must be numbered in all types of manuscript, including for Review.

This will be fixed in the type setting stage.

A 2nd section Methodology for literature searching must be developed.

Not sure what this is referring to.

Tabulated part is missing.

What table does the reviewer refer to?

Information is much too poor for a Review manuscript.

This is not a review, but an opinion piece.

Too many empty lines. What for?

This will be fixed in the type setting stage.

Less than 9 pages of real manuscript, in MDPI format which is 2/3 pages occupied with text cannot be considered a Review.

Again, this was an error in the submission process and has been rectified. 

Round 3

Reviewer 3 Report

In the key words you have selected "tau". Tau is a letter in the Greek alphabet not a keyword for medical topic.

The abbreviations have been not fixed. They must be explained also under each figure where the abbreviations were used.

L61. What is the novelty that this paper brings to the field or, at least the special aspects? Why you have made this study - reason for doing it, as in the literature are published tenths of thousands of papers in the topic.

Numbering the section is not something that must be done by the Publisher "This will be fixed in the type setting stage", but when submitting the manuscript.

How have you collected these data? Even you have not inserted the 2. Methodology section, in the main text it must be briefly mentioned the methodology you have performed for summarizing the info you have presented, the databases you have searched, etc.

Author Response

In the key words you have selected "tau". Tau is a letter in the Greek alphabet not a keyword for medical topic.

We have removed tau from the keywords.

The abbreviations have been not fixed. They must be explained also under each figure where the abbreviations were used.

We have gone through the manuscript and we believe to have fixed all the abbreviations.

L61. What is the novelty that this paper brings to the field or, at least the special aspects? Why you have made this study - reason for doing it, as in the literature are published tenths of thousands of papers in the topic.

We added a sentence that touch upon the rationale of the opinion piece.

Numbering the section is not something that must be done by the Publisher "This will be fixed in the type setting stage", but when submitting the manuscript.

This has been done.

How have you collected these data? Even you have not inserted the 2. Methodology section, in the main text it must be briefly mentioned the methodology you have performed for summarizing the info you have presented, the databases you have searched, etc.

As this is an opinion piece, we do not believe that a methodology section is appropriate here.  All statements, proposed ideas have been based on the published research and review articles which have been cited in the text. The papers are available on Pubmed and Google Scholar.